# From Traditional Typing to Genomic Precision: Whole-Genome Sequencing of *Listeria monocytogenes* Isolated from Refrigerated Foods in Chile

**DOI:** 10.3390/foods14020290

**Published:** 2025-01-16

**Authors:** Julio Parra-Flores, Beatriz Daza-Prieto, Pamela Chavarria, Miriam Troncoso, Anna Stöger, Guillermo Figueroa, Jetsi Mancilla-Rojano, Ariadnna Cruz-Córdova, Aleksandra Martinovic, Werner Ruppitsch

**Affiliations:** 1Department of Nutrition and Public Health, Universidad del Bío-Bío, Chillán 3780000, Chile; pchavarria@ubiobio.cl; 2Institute of Medical Microbiology and Hygiene, Austrian Agency for Health and Food Safety, 1090 Vienna, Austria; beatriz.daza-prieto@ages.at (B.D.-P.); anna.stoeger@ages.at (A.S.); 3Fundación Instituto Profesional Duoc UC, Santiago 8240000, Chile; troncoso56@gmail.com; 4Microbiology and Probiotics Laboratory, Institute of Nutrition and Food Technology, Universidad de Chile, Santiago 7830490, Chile; gfiguerog@gmail.com; 5Immunochemistry Research Laboratory, Hospital Infantil de México Federico Gómez, Mexico City 06720, Mexico; mancillajetsi@gmail.com (J.M.-R.); ariadnnacruz@yahoo.com.mx (A.C.-C.); 6Faculty of Food Technology, Food Safety and Ecology, University of Donja Gorica, 81000 Podgorica, Montenegro; aleksandra.martinovic@udg.edu.me

**Keywords:** *Listeria monocytogenes*, whole-genome sequencing, antimicrobial resistance genes, virulence genes

## Abstract

Ready-to-eat (RTE) foods are the most common sources of *Listeria monocytogenes* transmission. Whole-genome sequencing (WGS) enhances the investigation of foodborne outbreaks by enabling the tracking of pathogen sources and the prediction of genetic traits related to virulence, stress, and antimicrobial resistance, which benefit food safety management. The aim of this study was to evaluate the efficacy of WGS in the typing of 16 *L. monocytogenes* strains isolated from refrigerated foods in Chile, highlighting its advantages in pathogen identification and the improvement of epidemiological surveillance and food safety. Using cgMLST, a cluster was identified comprising 2 strains with zero allele differences among the 16 strains evaluated. Ninety-four percent of the isolates (15/16) were serotype 1/2b, and 88% of them (14/16) were ST5. All strains shared identical virulence genes related to adhesion (*ami*, *iap*, *lapB*), stress resistance (*clpCEP*), invasion (*aut*, *iapcwhA*, *inlAB*, *lpeA*), toxin production (*hly*), and intracellular regulation (*prfA*), with only 13 strains exhibiting the *bcrBC* and *qacJ* gene, which confer resistance to quaternary ammonium. The pCFSAN010068_01 plasmids were prevalent, and insertion sequences (ISLs) and composite transposons (cns) were detected in 87.5% of the strains. The presence of various antibiotic resistance genes, along with resistance to thermal shocks and disinfectants, may provide *L. monocytogenes* ST5 strains with enhanced environmental resistance to the hygiene treatments used in the studied food production plant.

## 1. Introduction

Ready-to-eat (RTE) foods are defined as foods consumed in their raw state or manipulated, processed, mixed, cooked, prepared, and consumed without requiring further handling [1]. These products are increasingly in demand due to the fast-paced lifestyles where time for food preparation is limited and becoming less healthy. While RTE foods provide a convenient solution for everyday dietary needs, they are susceptible to potential contamination by various agents, with biological agents, particularly pathogens, standing out among them, posing serious health risks to consumers [2].

Among these risks, contamination with pathogenic microorganisms such as *Salmonella* spp., pathogenic *Escherichia coli*, and *Listeria monocytogenes* are identified [3]. *L*. *monocytogenes*, a Gram-positive, facultative anaerobic bacterium, exhibits resilience in various foods. In Europe and North America, invasive listeriosis affects 0.3–0.6 persons/100,000 annually. This bacterium is responsible for listeriosis, a disease manifesting through abortions, septicemia, meningitis, and, in severe cases, death [4]. Between 2005 and 2020, a total of 3628 cases of listeriosis were reported across 127 outbreaks associated with RTE foods. Of these outbreaks, 54% occurred in the European region and 38% in the Americas. Additionally, 31% of the outbreaks were linked to RTE meat products, 28% to RTE dairy products, 13% to fresh or minimally processed RTE fruit and vegetable products, 12% to fish and seafood products, and the remainder to foods containing multiple ingredients [5]. In Chile, two significant listeriosis outbreaks were reported between 2008 and 2009, with 165 and 73 cases, respectively. These outbreaks were associated with the consumption of goat cheese, sausages, and other meat products. The most frequently identified serotypes were 4b (CC1) and 1/2a (CC9) [6].

While listeriosis can impact anyone, the highest-risk group comprises pregnant women, children, older adults, and immunocompromised individuals or those with autoimmune diseases [7]. The defining characteristic in this high-risk group is a compromised or immature immune system (in the case of children), rendering them more susceptible to this disease. This microorganism resides in the gastrointestinal systems of animals and can cause illness in them. Additionally, it can migrate and contaminate various foods due to improper manufacturing practices and inadequate hygiene [8]. This particularly affects unpasteurized dairy products, raw meats, and non-animal origin food such as frozen vegetables and fruits. Failure to undergo proper heat treatments in these foods can be detrimental to consumer health. Foods most commonly associated with outbreaks nationally and internationally include cheeses (Camembert, brie), cured meats, and fish [8,9]. Outbreaks and cases caused by *L. monocytogenes* are closely linked to serotypes 1/2a, 1/2b, and 4b, accounting for 95% of human listeriosis cases. Additionally, the severity is associated with multiple virulence factors, such as the ability to form biofilms and resistance to antibiotics, especially beta-lactams (penicillin, ampicillin, vancomycin) [10,11].

Monitoring pathogenic microorganisms in food using traditional culture methods or viable counts provides direct risk evidence but can take up to 7 days due to the low concentration and stress conditions of these bacteria. Recent research in foodborne disease epidemiology has utilized a range of techniques from immunology to genetics [12,13], to help identify the sources of contamination or infection during outbreaks [12]. These techniques are crucial for active epidemiological surveillance.

PulseNet historically used pulsed-field gel electrophoresis (PFGE) to characterize pathogen isolates in foodborne outbreaks, establishing common etiological agents among seemingly unrelated outbreaks [14,15]. However, PFGE is labor-intensive, prone to false positives, and challenging to reproduce across laboratories [16]. Currently, WGS is the standard method for molecular typing and epidemiological surveillance [17]. WGS technologies offer retrospective compatibility with datasets obtained through traditional methods by extracting comprehensive genomic information. This includes determining the serotype, the classical multilocus sequence type (MLST), the core genome MLST (cgMLST), virulence factors, toxigenicity, antibiotic resistance, mobile genetic elements (MGEs), and plasmids of an isolate in a single experiment. The characterization of microorganisms using WGS, whether through single nucleotide variants (SNVs) or gene-by-gene allelic profiling, is now the most powerful diagnostic tool for detailed microorganism analysis [18,19,20,21,22,23]. The aim of this study was to evaluate the efficacy of WGS in the typing of 16 *L. monocytogenes* strains isolated from refrigerated foods, highlighting its advantages in pathogen identification and the improvement of epidemiological surveillance and food safety.

## 2. Materials and Methods

### 2.1. Sampling and Identification of Isolates

After detecting a strain of *L. monocytogenes* in a refrigerated RTE food, a comprehensive study was conducted at the RTE production plant located in Santiago, Chile (33°26′16″ S, 70°39′01″ W), to identify specific sources of contamination. A two-class sampling plan was applied with the parameters n = 5; c = 0; m = 0; and M = 0 where n is the number of sample units drawn; c: the maximum allowable number of positive results; m: the unacceptable tested samples; and M: the boundary between marginally acceptable counts and unacceptable counts. A total of 216 samples were analyzed from various areas of the plant, including surfaces (floors, grates, and drains), equipment (worktables, transport cart wheels, mixing paddle and preparation tanks), raw materials, and finished products. The ISO 11290-1:1996/AMD 1:2004 method was used for isolation, and the BAX^®^ System (DuPont, Qualicon, Wilmington, DE, USA) for confirmation. Additionally, the identification of *L. monocytogenes* was confirmed by PCR with primers from the *iap* gene (MonoA 5′CAAACTGCTAACACAGCTACT3′ and MonoB 5′GCACTTGAATTGCTGTTATTG3′) [24].

Sixteen strains of *L. monocytogenes* were identified during this study, included with the originally detected strain, all of which were stored at −80 °C for future investigations.

### 2.2. PFGE

PFGE was carried out following the Center for Disease Control and Prevention (CDC)’s standardized PulseNet protocol for *L. monocytogenes* https://pulsenetinternational.org/assets/PulseNet/uploads/pfge/PNL04_ListeriaPFGEProtocol.pdf (accessed 1 December 2024) with ApaI (Promega Corporation, Madison, WI, USA) as the restriction endonuclease. The PFGE patterns were examined using the Gel ComparII Software (Bionumerics v5.0, Applied Maths NV Keistraat, Sint-Martens-Latem, Belgium), which utilizes the standard strain *Salmonella enterica* serovar Braenderup H9812, loaded into three lanes of each gel, to standardize the images. Additionally, the reference strain *L. monocytogenes* ATCC 19115 was employed as a control. Comparison and the UPGMA dendrogram (unweighted pair group method with averages) analysis of the PFGE patterns were conducted using the Dice coefficient with a 1.5% tolerance range.

### 2.3. Reidentification of Listeria monocytogenes Isolates

After initial identification using BAX system, re-identification was conducted using matrix-assisted laser desorption ionization−time-of-flight mass spectrometry (MALDI-TOF-MS) (Bruker, Billerica, MA, USA) using MBT Compass IVD software 4.1.60 (Bruker), as described by Halbedel et al. [4].

### 2.4. WGS

Bacterial isolates were grown on blood agar plates supplemented with 5% sheep blood (COS) (Biomerieux, Vienna, Austria) at 37 °C/24  h/aerobiosis. High-molecular-weight DNA was isolated from overnight cultures using the KingFisher Apex System (Thermofisher, Vienna, Austria) with the Omega Mag-Bind Bacterial DNA 96 Kit (Ref. M2350-01) following the manufacturer’s instruction for Gram-positive bacteria with the following modifications: 2 hours of lysis with 20 μL of lysozyme and 1 h of incubation with 30 µL of proteinase K (20 mg/mL). DNA was quantified using Dropsense 16 (Trinean NV/SA, Gentbrugge, Belgium) and the Qubit Flex Fluorometer (Thermofisher, Vienna, Austria) with 1x dsDNA High Sensitivity (HS) kit (Thermofisher, Vienna, Austria). Genomic libraries were prepared using the DNA Illumina Prep kit (Illumina, San Diego, CA, USA) according to the manufacturer’s instructions. Paired-end sequencing was performed on a NextSeq 2000 instrument (Illumina, San Diego, CA, USA) with a read length of 2 × 301 bp (Illumina, San Diego, CA, USA) aiming for a minimum coverage of 30-fold. Raw reads underwent quality control with FastQC v0.11.9, and adapter sequences were removed. Additionally, the last 10 bp of each sequence and sequences with quality scores below 20 were trimmed using Trimmomatic v0.36 [25]. Subsequently, the raw reads were de novo assembled using SPAdes v3.11.1 [26], where the contigs were filtered based on a minimum coverage of 5x and a minimum length of 200 bp using SeqSphere+ software v10.0 (Ridom GmbH, Würzburg, Germany) [27].

The annotation of strains was performed using Proksee v1.3.0 [28] and Prokka v1.1.1 [29].

### 2.5. Genomic Identification, Serotype, ST, and cgMLST of Listeria monocytogenes

The 16 *L. monocytogenes* strains were confirmed using ribosomal multilocus sequence typing (rMLST) through a method that examines the variation in the 53 genes responsible for encoding the protein subunits of the bacterial ribosome (*rps* genes), which is accessible at https://pubmlst.org/species-id (accessed on 1 October 2024) [30]. From the WGS analysis *of L. monocytogenes* strains, serotypes were determined through the sequence-specific extraction of targets using the *L. monocytogenes* 5-plex PCR Serogroup task templates in SeqSphere+ v. 10.0.1 (2024-05) software. This utilized fragments from five DNA regions (*lmo118*, *lmo0737*, *ORF2110*, *ORF2829*, and *prs* serving as an internal amplification control), as previously described by Doumith et al. and Lee et al. [31,32].

STs were determined using Task templates for available MLST schemes in SeqSphere+ v. 10.0.1 (2024-05) software. ST confirmation in the strains involved fragments from the seven housekeeping genes *abcZ*, *bglA*, *cat*, *dapE*, *dat*, *Idh*, and *ihkA* [33,34], along with profiles from the Institut Pasteur MLST Listeria database (https://bigsdb.pasteur.fr/cgi-bin/bigsdb/bigsdb.pl?db=pubmlst_listeria_seqdef; accessed 1 December 2024).

The cgMLST analysis was conducted using the profile of 1701 loci of cgMLST complex types (CTs) [35] using Task templates for SeqSphere+ v. 10.0.1 (2024-05). A cgMLST cluster was defined as a group of strains with fewer than 10 different alleles among the studied isolates. Phylogenetic trees were generated using SeqSphere+ in a mode that ignored pairwise missing values and employed an unweighted pair group method with the arithmetic mean [4].

### 2.6. Detection of Virulence and Antibiotic Resistance Genes

Virulence genes were identified using the VFDB v.2.0 feature task template in SeqSphere+ for WGS data [36]. For the target scanning procedure, thresholds were set at a required identity ≥ 90% with the reference sequence and an aligned reference sequence ≥ 99%.

The Comprehensive Antibiotic Resistance Database (CARD) was employed with the default “perfect” and “strict” settings for the sequence analysis of antimicrobial resistance genes [37]. Additionally, the Task Template AMRFinderPlus v.3.11.26, available in the Ridom SeqSphere+ 10.0.1 software, was utilized with the EXACT method at the 100% setting, along with the BLAST alignment of protein sequences against the AMRFinderPlus database [38].

### 2.7. Plasmids and Mobile Genetic Element Detection

The detection of plasmids was carried out using the MOB-suite tool v3.1.8 [39] integrated in Ridom SeqSphere v10.01 and the CGE MobileElementFinder v1.1.2 [40] (accessed on 1 October 2024).

### 2.8. Bioinformatic Search for CRISPR/Cas Systems

The search and characterization of CRISPR arrays and their association with Cas proteins were determined using CRISPRDetect [41] (available at http://crispr.otago.ac.nz/CRISPRDetect/predict_crispr_array.html and http://www.microbiome-bigdata.com/CRISPRminer accessed on 1 December 2024) with the following parameters: a repeat sequence length of 11 bp to 55 bp, a spacer length of 25 to 60 bp, a spacer size relative to the repeat sequence length of 0.6 to 2.5, and a maximum percentage of similarity between spacers of 67%.

The CRISPRTarget program was used to determine the PAM sequences (Protospacer Adjacent Motifs) and the genetic information associated with each of the spacer sequences in the identified arrays.

## 3. Results

### 3.1. Isolation and Primary Species Identification of Listeria monocytogenes

Presumptive strains of *L. monocytogenes* were identified in 16 out of the 216 analyzed samples, representing 4.4% of the total. The highest positivity rate was observed in surfaces and equipment, at 11.6% (12/103), followed by finished product samples, at 3.8% (4/105). No *L. monocytogenes* strains were found in raw materials (Table 1). All strains were initially confirmed as *L. monocytogenes* through amplification of the *iap* gene using PCR.

### 3.2. PFGE

The 16 strains were characterized using PFGE, resulting in a correlation coefficient of 0.90406 (Figure 1). Additionally, one group of finished products (510085-23, 511472, and 511475-24) was identified, containing six closely related strains found in surfaces and equipment (510086-23, 510087-23, 510088-23, 510090-23, 510231-23, and 510250-23) (Group A). In contrast, the other two groups (B and C) exhibited greater variability and were not related to the strains isolated from finished products (Figure 1).

### 3.3. Genomic Identification, Serotype, ST, and cgMLST of Listeria monocytogenes

All presumptive strains (100%) were confirmed as *L. monocytogenes* using MALDI-TOF MS and rMLST. The dominant serotype was 1/2b, found in 93% of the strains (15/16), while the remaining serotype was 4b. Regarding sequence types, 14 strains were identified as ST5 (CC5), one strain as ST2349 (CC5), and the single serotype 4b strain as ST145 (CC2) (Table 2).

When conducting WGS analysis of the strains using cgMLST, a total of 12 complex types were identified, with the most frequent being CT9170 in four strains and CT18381 in two strains, both corresponding to samples from surfaces and equipment. The four strains isolated from finished products exhibited different complex types. Only one cluster was observed, consisting of 2 identical strains sharing the same CT9170, while the remaining 14 strains, some with similar complex types, were found to be unrelated to each other (Figure 2).

### 3.4. Detection of Virulence and Antibiotic Resistance Genes

All ST5 *L. monocytogenes* strains present the same virulence genes associated with adhesion (*ami*, *iap*, *lapB*), stress resistance (*clpCEP*), invasion (*aut*, *iapcwhA*, *inlAB*, *lpeA*), toxin production (*hly*), and intracellular regulation (*prfA*). In contrast, the ST2349 strain lacks *ami*, *aut*, and *labp* but is the only strain that contains the aggregation gene *actA*.

Additionally, 100% of the analyzed strains possess the resistance gene to fosfomycin (*fosx*) and lincosamide (*vga).* However, the *bcrBC* gene, which confers resistance to quaternary ammonium compounds (QACs) and the *qacJ* gene associated with the efflux mechanisms conferring resistance to disinfecting agents and antiseptics, was identified in only 13 strains, excluding strains 51088-23, 510090-23, and 511475-23 (Table 3).

### 3.5. Plasmids and Mobile Genetic Element Detection

A total of 87.5% (14/16) of the strains exhibited plasmids, with the most common being CP014251 (pCFSAN010068_01), which was detected in 13 strains and contained various MGEs such as insertion sequences (ISLs) and composite transposons (cns). In addition, plasmid LR134399 was identified in two strains, with a size exceeding 400,000 bp and containing only an ISL as the MGE. Furthermore, the isolate 511471-24 carried plasmid CP014251 (Table 4).

### 3.6. Bioinformatic Search for CRISPR/Cas Systems

The genome analysis revealed the presence of potential CRISPR/Cas systems in 100% (*n* = 16/16) of the genomes, with either arrays or Cas genes observed, but never both simultaneously. In the *L. monocyogenes* isolates, 82% (*n* = 13/16) of the genomes displayed arrays characterized by up to eight repeat sequences and seven spacers, but in different positions of the genome (Appendix A).

The repeat sequences and associated Cas genes allowed the CRISPRTarget program to determine that 88% (*n* = 14/16) of probable CRISPR systems could be associated with the type I-B. Regarding the associated Cas genes, the csa3 gene was identified in all genomes (Appendix A). Spacer analysis using CRISPRTarget linked these sequences, in seven *Listeria* genomes, to bacteriophage characteristics of the same genus, all associated with the bacteriophage *Listeria* phage LM 4-11-1 and PHAGE_Lister_A118 (Appendix A). Additionally, the TAC or CAT sequence was identified as the PAM sequence (Protospacer Adjacent Motif).

## 4. Discussion

The presence of *L. monocytogenes* represents a serious risk to the food industry and public health due to several factors that allow certain strains to persist in food processing environments and become a frequent source of contamination [42].

In this study, 16 strains of *L. monocytogenes* were identified, 12 of which were isolated from surface samples in food production areas, and 4 from finished food products. Initially, the isolates were molecularly characterized using PFGE, concluding that the contamination was due to the presence of a unique profile, indicating that the pathogen had become established in the plant, particularly in favorable ecosystems such as drains, where it can form biofilms [43]. Furthermore, PFGE allowed the conclusion that the main group of strains isolated in this study was not related to virulent clones isolated during the extensive outbreaks in Chile in 2009 and 2012. Subsequently, WGS of all *L. monocytogenes* isolates allowed for a more precise and comprehensive characterization of the strains. For this reason, monitoring programs for *L. monocytogenes* that use molecular subtyping are essential tools for identifying persistent isolates throughout the food chain [44]. Moreover, WGS provides the maximum resolution for identifying, characterizing, and subtyping this pathogen, significantly outperforming PFGE in foodborne outbreak surveillance plans [45].

Based on WGS characterization, 15 isolates were primarily grouped into serogroup IIb (serotype 1/2b), and one isolate belonged to serogroup IVb (serotype 4b). In Chile, serotypes 1/2a, 1/2b, and 4b are the most frequently detected in foods and clinical cases of listeriosis [46]. A study conducted in England, focusing on frozen vegetables, found that the most common serotype among *L. monocytogenes* isolates was serotype 1/2a, although serotypes 1/2b and 4b were also identified [47]. Similarly, Daza et al. [48] reported in their study on the food chain in Montenegro that *L. monocytogenes* isolates came from 80% meat, 16% dairy products, 0.7% ready-to-eat fish, and 0.7% frozen vegetables, with the most prevalent serogroup being IIa, followed by IIc. These results are consistent with those obtained by Pyz-Lukazik et al. [49] in artisan cheeses in Poland, and they differ from those of our study, which focused on refrigerated ready-to-eat (RTE) foods. In this context, serotypes 1/2b and 4b are more frequently associated with human listeriosis, while serotype 1/2a is most commonly isolated from foods and food processing environments [50].

Ruppitsch et al. [35] developed a highly representative cgMLST typing scheme for the WGS-based analysis of *L. monocytogenes*, expanding the classical MLST approach to a genome-wide gene-by-gene comparison. This scheme demonstrated a high discriminatory power and strong concordance with previous findings in various outbreak scenarios [51]. It enabled the clustering of outbreak-related isolates with differences of ≤7 alleles, as well as the clear separation of unrelated isolates [52].

In our study, using the classical MLST scheme [33], *L. monocytogenes* ST5 (CC5) was identified as the most prevalent ST. Using the cgMLST gene-by-gene analysis scheme [35], we identified only one cluster comprising two strains (510225-23 and 510250-23) with zero allelic difference, which were isolated from the factory floor and a production surface, respectively. The remaining ST5 strains exhibited between 11 and 38 allele differences, while the ST145 (CC2) strain had 1666 allele differences, indicating that they were not related strains. Several studies reported that *L. monocytogenes* ST5 strains are frequent and persistent contaminants in RTE food processing environments [53,54]. Additionally, they identified *L. monocytogenes* ST5 isolates with the same pulsotype in both processing environments and final products from the evaluated plants. Furthermore, analysis through cgMLST revealed up to nine allelic differences, with the closest pairwise differences observed among these ST5 isolates. In this context, the use of the classical MLST approach to a genome-wide gene-by-gene comparison (cgMLST) has facilitated the establishment of well-defined, open-access databases based on core or whole-genome MLST schemes. These databases significantly enhance the tracking of *L. monocytogenes* sources and the investigation of multinational outbreaks, enabling more efficient collaboration between public health laboratories globally and supporting the adoption of preventive measures to protect public health [51]. In 2002, a study on ready-to-eat foods in Chile reported the isolation of an ST5 strain (CT8052) from cooked shrimp, which was unrelated to this study [55].

Several studies on the pathogenic characteristics and persistence of *L. monocytogenes*, as well as the virulence genes of its strains, serve as key tools for assessing the risk of infections associated with isolates from food sources [56]. The virulence traits of *L. monocytogenes* are determined by numerous genes and proteins, some of which are grouped into genomic and pathogenicity islands that are closely linked to the pathogen’s infectious life cycle [57]. In our study, we detected genes from the LIPI-1 group in all the strains evaluated, which are involved in the intracellular infection cycle of *L. monocytogenes*; 100% of the analyzed strains carried the genes *hlyA*, *prfA*, *plcB*, and *inlA*. Additionally, the strain 510090-23 (ST145-CC2) was found to harbor the genomic island LGI-2, which includes genes for arsenic resistance (arsenic resistance cassette *arsABDD2R*) and cadmium resistance (*cadAC*). It is worth noting that LGI-2 has been primarily identified in *L. monocytogenes* serotype 4b strains, including hypervirulent clones belonging to clonal complexes CC1 and CC2 [58,59,60,61]. Therefore, the strain identified in our study could potentially be a risk to consumers of these food products.

In our study, all analyzed *Listeria monocytogenes* strains presented the *fosX* (fosfomycin) and *vgaG* (lincosamide) resistance genes. Fosfomycin is a broad-spectrum bactericidal antibiotic effective against both Gram-positive and Gram-negative bacteria, including multidrug-resistant strains. Its mechanism of action provides a synergistic effect with numerous antimicrobials and makes cross-resistance exceptionally rare, which makes it a highly attractive antibiotic, particularly for the treatment of infections caused by multidrug-resistant bacteria [62]. Another gene identified in all analyzed strains was the *vgaG* gene, which encodes resistance to lincosamide. Lincosamide is an antibiotic that was discovered in 1962 through the purification of an actinomycete, leading to the development of two marketed molecules: lincomycin and clindamycin. In practice, clindamycin is the reference lincosamide, widely cited in therapeutic recommendations [63]. These antibiotics act by inhibiting the transpeptidation necessary for bacterial protein synthesis, exhibiting bacteriostatic activity [64]. Mota et al. [65] suggested that the presence of resistance genes for fosfomycin and lincosamide is considered a form of natural resistance or inherent immunity of the bacterium.

The increasing tolerance to disinfectants used in industrial food processing plants represents a critical factor in the survival of microorganisms such as *L. monocytogenes*. For this reason, operational cleaning and disinfection procedures commonly involve the use of QACs [66,67]. Several studies identified the presence of genes associated with QAC resistance, with *norB*, the *bcrABC* cassette, *emrE*, and *Tn6188* as the most frequently reported [68]. In our study, we detected the presence of *bcrBC and qacJ* genes in 85% (12/14) of the ST5 (CC5) isolates and in the ST2349 (CC5) isolate. Furthermore, 100% of the isolates carried the *norB* gene, which encodes a quinolone resistance efflux pump. Some authors reported that the *bcrABC* cassette is conserved in the chromosome and confers tolerance to quaternary ammonium compounds [69], which aligns with the findings observed in our isolates. Moreover, the presence of the *qacJ* gene in 13 isolates of *L. monocytogenes*, which is associated with efflux mechanisms that confer resistance to disinfectants and antiseptics, poses an additional challenge to eradication efforts, particularly in food production environments due to the persistence of this pathogen. It has also been shown that L. monocytogenes strains capable of adapting to biocides, specifically QACs, may be associated with the development of resistance to the antibiotic ciprofloxacin [70].

The plasmid CP014251 (*pCFSAN010068_01*) was prevalent among the strains analyzed in our study. This plasmid had previously been reported in the outbreak associated with Hispanic-style cheese in Maryland in 2013 [71]. The plasmid belongs to the *repA* family and is characterized by the presence of insertion sequences, primarily from the IS3 and IS6 families, as well as recombinase genes [72]. Additionally, two of the strains analyzed carried the plasmid LR134399 (*NCTC7974*), a high-molecular-weight plasmid that harbors the gene for lincosamide resistance and was originally isolated from a clinical strain of *L. monocytogenes* [73].

The CRISPR/Cas systems allow bacteria to acquire exogenous genetic material from bacteriophages and plasmids, thereby providing protection against bacteriophage infections [74]. In the *L. monocytogenes* strains studied, the repeated sequences and associated *Cas* genes were identified as belonging to type I-B and I-A systems. Furthermore, these repeated sequences and spacers were only associated with the *Cas3* gene, which is involved in the integration and processing of acquired information [75]. Regarding the spacer sequences, these provide a historical record of invasive elements that the bacteria have been exposed to. For the arrays identified in this study, the spacers were associated with bacteriophage sequences that specifically infect the *Listeria* genus. In the analyzed strains, the presence of Listeria phage A118 was observed, a temperate bacteriophage specific to *Listeria monocytogenes* serovar 1/2 strains, which are often implicated in foodborne disease outbreaks [76]. It has been proposed that bacteria with CRISPR arrays but lacking *Cas* genes may incorporate information from bacteriophages into their genome as prophages. On the other hand, bacteria lacking a CRISPR/Cas system can be infected by bacteriophages, acquiring genes related to resistance and virulence. This enhances their adaptive capacity to the environment and, as seen in these *L. monocytogenes* strains, poses a significant health risk [77].

## 5. Conclusions

The genomic analysis of the *L. monocytogenes* strains studied revealed the presence of various antibiotic resistance genes, as well as resistance to thermal shocks and disinfectants. These traits may confer acquired environmental resistance to the hygiene treatments used in the food production plant under study, thereby promoting the persistence of these *L. monocytogenes* ST5 strains. Consequently, the use of WGS provides valuable information that allows for the more precise targeting of preventive sanitization programs, thus ensuring food safety and contributing to a healthier and safer diet. In this context, WGS provides greater resolution and accuracy in strain identification compared to traditional typing, in addition to a genomic characterization, which enhances the understanding of evolutionary relationships and responses in epidemiological surveillance. Accurate strain identification is key in the management of pathogens such as *L. monocytogenes* in food and processing facilities, as it is essential for ensuring food safety and protecting public health.

## Figures and Tables

**Figure 1 foods-14-00290-f001:**
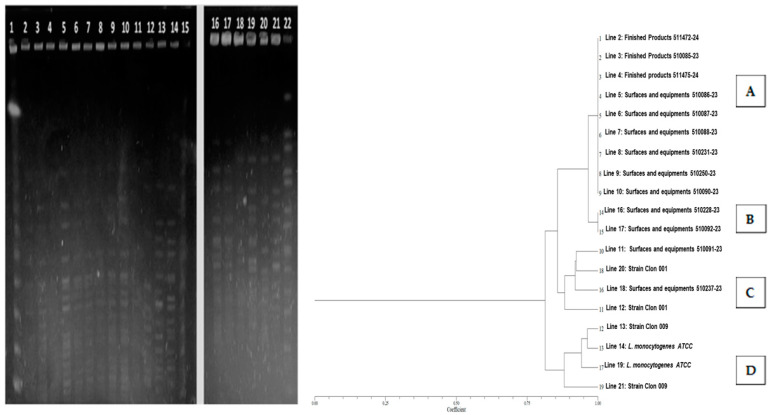
PFGE of *L. monocytogenes* strains isolated from surfaces and equipment, as well as finished products. In Group A, six closely related strains from surfaces and production equipment are observed alongside three strains isolated from finished products, all displaying unique profiles. Additionally, two other isolates from surfaces and equipment (510092-23 and 510228-23) are noted as a subgroup within Group A. There is no observed relationship with persistent *L. monocytogenes* strains Clon 1 and Clon 9 isolated from clinical cases. The *L monocytogenes* strains of Groups B–D are not related.

**Figure 2 foods-14-00290-f002:**
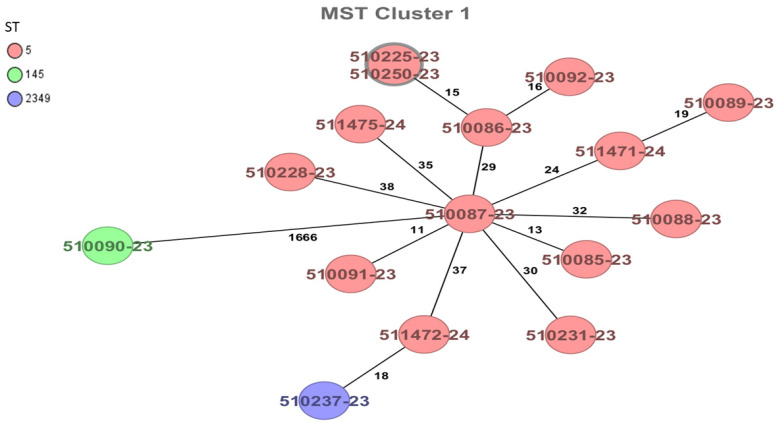
The minimum spanning tree (MST) of sixteen *Listeria monocytogenes* strains isolated from refrigerated foods and surfaces and equipment. The MST is calculated based on the defined cgMLST scheme [33], which comprises 1701 target genes for *Listeria monocytogenes.* Isolates are represented as colored circles according to the classical MLST, with black numbers indicating the allelic differences between isolates. Two isolates with closely related genotypes are marked as clusters.

**Table 1 foods-14-00290-t001:** Positivity of *Listeria monocytogenes* by sample source.

Source	Number of Samples *n*	Positivity *n* (%)
Surfaces and equipment	103	12 (11.6)
Raw material samples	18	0 (0)
Finished product samples	105	4 (3.8)
Total	216	16 (4.4)

**Table 2 foods-14-00290-t002:** Identification of strains using MALDI-TOF MS), rMLST and cgMLST based on WGS.

Sample ID	Food	MALDI-TOF MS	rMLST	ST	CC	CT	Serotype
510085-23	Finished product	*L. monocytogenes*	*L. monocytogenes*	5	CC5	9152	1/2b
510086-23	Surfaces and equipment (floor)	*L. monocytogenes*	*L. monocytogenes*	5	CC5	9170	1/2b
510087-23	Surfaces and equipment (worktable)	*L. monocytogenes*	*L. monocytogenes*	5	CC5	18381	1/2b
510088-23	Surfaces and equipment (preparation tank)	*L. monocytogenes*	*L. monocytogenes*	5	CC5	18382	1/2b
510089-23	Surfaces and equipment (drain)	*L. monocytogenes*	*L. monocytogenes*	5	CC5	18384	1/2b
510091-23	Surfaces and equipment (worktable)	*L. monocytogenes*	*L. monocytogenes*	5	CC5	18381	1/2b
510092-23	Surfaces and equipment (transport cart wheel)	*L. monocytogenes*	*L. monocytogenes*	5	CC5	9170	1/2b
511471-24	Finished product	*L. monocytogenes*	*L. monocytogenes*	5	CC5	10192	1/2b
510225-23	Surfaces and equipment (preparation tank)	*L. monocytogenes*	*L. monocytogenes*	5	CC5	9170	1/2b
510228-23	Surfaces and equipment (mixing paddle)	*L. monocytogenes*	*L. monocytogenes*	5	CC5	18386	1/2b
511472-24	Finished product	*L. monocytogenes*	*L. monocytogenes*	5	CC5	21278	1/2b
510231-23	Surfaces and equipment (grate)	*L. monocytogenes*	*L. monocytogenes*	5	CC5	8061	1/2b
510250-23	Surfaces and equipment (floor)	*L. monocytogenes*	*L. monocytogenes*	5	CC5	9170	1/2b
510237-23	Surfaces and equipment (grate)	*L. monocytogenes*	*L. monocytogenes*	2349	CC5	18391	1/2b
510090-23	Surfaces and equipment (worktable)	*L. monocytogenes*	*L. monocytogenes*	145	CC2	66	4b
511475-24	Finished product	*L. monocytogenes*	*L. monocytogenes*	5	CC5	21280	1/2b

**Table 3 foods-14-00290-t003:** Antibiotic resistance and virulence genes in *Listeria monocytogenes* strains isolated according to sequence type (ST).

ST	CC	Virulence Genes	Resistance Genes
145	CC2	*bsh*, *clpC*, *clpE*, *clpP*, *fbpA*, *hly*, *hpt*, *iap*/*cwhA*, *inlA*, *inlB*, *inlC*, *inlP*, *lap*, *lntA*, *lpeA*, *lplA1*, *lspA*, *mpl*, *oatA*, *pdgA*, *plcA*, *plcB*, *prfA*, *prsA2*	*fosX*, *vga(G)*
2349	CC5	*ami*, *aut*, *bsh*, *clpC*, *clpE*, *clpP*, *fbpA*, *gtcA*, *hly*, *hpt*, *iap/cwhA*, *inlA*, *inlB*, *inl*, *inlK*, *inlP*, *lap*, *lapB*, *lntA*, *lpeA*, *lplA1*, *lspA*, *mpl*, *oatA*, *pdgA*, *plcA*, *plcB*, *prfA*, *prsA2*	*bcrB*/*bcrC*, *fosX*, *vga(G)*
5	CC5	*ami*, *aut*, *bsh*, *clpC*, *clpE*, *clpP*, *fbpA*, *gtcA*, *hly*, *hpt*, *iap/cwhA*, *inlA*, *inlB*, *inlC*, *inlK*, *inlP*, *lap*, *lapB*, *lntA*, *lpeA*, *lplA1*, *lspA*, *mpl*, *oatA*, *pdgA*, *plcA*, *plcB*, *prfA*, *prsA2*	*bcrB*/*bcrC*, *qacJ*, *fosX*, *vga(G)*

ST: sequence type; CC: clonal complex.

**Table 4 foods-14-00290-t004:** Presence of plasmids and mobile genetic elements in *Listeria monocytogenes* strains.

ID Strain	ST	Plasmid	Plasmid Accession Number	Size (Kb)	Mobile Genetic Elements
510085-23	5	P40.3_510085	CP014251	40,325	cn_8427_ISLmo3, ISLmo3, cn_17195_ISLmo3, ISLmo4, ISLmo5, ISLmo19
510086-23	5	P68.0_510086	CP014251	68,042	cn_8427_ISLmo3, ISLmo3, cn_17195_ISLmo3, ISLmo4, cn_12275_ISLmo3, ISLmo19, cn_13026_ISLmo19, ISLmo8, ISLmo4, ISLmo5
510087-23	5	p34.8_510087	CP014251	34,816	cn_8427_ISLmo3, ISLmo3, ISLmo4, ISLmo5, ISLmo19,
510088-23	5	No found			
510089-23	5	p34.8_510089	CP014251	34,816	ISLmo3, ISLmo4, ISLmo5, ISLmo19,
510091-23	5	p34.8_510091	CP014251	34,816	ISLmo3, ISLmo4, ISLmo5, ISLmo19, Cn_8427_ISLmo3
510092-23	5	P69.1_510092	CP014251	69,077	ISLmo5, cn_13026_ISLmo4, ISLmo4, cn_12275_ISLmo19, ISLmo8, cn_17195_ISLmo19, ISLmo19, cn_17195_ISLmo3, cn_8427_ISLmo3
511471-24	5	P406.7_511471 P50.7_511471	LR134399 CP014251	406,655 50,672	ISLmo3, ISLm19, ISLmo5, ISLmo4
510225-23	5	P69.1_510092	CP014251	69,077	ISLmo5, cn_13026_ISLmo4, ISLmo4, cn_12275_ISLmo19, ISLmo8, cn_17195_ISLmo19, ISLmo19, cn_17195_ISLmo3, cn_8427_ISLmo3, ISLmo3
510228-23	5	P34.6_510228	CP014251	34,626	cn_8427_ISLmo3, ISLmo3, ISLmo4, ISLmo5, ISLmo19, ISLmo8
511472-24	5	P51.1_511472	CP014251	51,100	cn_7697_ISLmo3, ISLmo3, ISLmo4, ISLmo5, ISLmo19, ISLmo8
510231-23	5	P37.2_510231	CP014251		cn_8427_ISLmo3, ISLmo3, ISLmo4, ISLmo5, ISLmo19, ISLmo8
510250-23	5	P69.1_510092	CP014251	69,077	cn_8427_ISLmo3, ISLmo3, cn_17195_ISLmo3, ISLmo4, cn_12275_ISLmo3, ISLmo19, cn_13026_ISLmo19, ISLmo8, ISLmo4, ISLmo5
510237-23	2349	P40.3_510237	CP014251	40,325	cn_8427_ISLmo3, ISLmo3, ISLmo4, ISLmo5, ISLmo19, ISLmo8
510090-23	145	No found			
511475-24	5	P426_511475	LR134399	425,974	Not found

## Data Availability

The original contributions presented in the study are included in the article/Appendix A, further inquiries can be directed to the corresponding author.

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
