# Peer review of "From Traditional Typing to Genomic Precision: Whole-Genome Sequencing of *Listeria monocytogenes* Isolated from Refrigerated Foods in Chile"

_foods, 2025, doi:10.3390/foods14020290_

Round 1
Reviewer 1 Report
Comments and Suggestions for Authors
This study investigates Listeria monocytogenes isolates collected from refrigerated ready-to-eat (RTE) foods and processing environments in Chile, employing whole genome sequencing (WGS) to characterize their genetic profiles. The analysis reveals key insights into the serotypes, sequence types, virulence factors, antimicrobial resistance genes, and mobile genetic elements present in the isolates. There are some problems that should be addressed in its current form.
The manuscript should highlight in the title, abstract, and relevant sections that the isolates were obtained from Chile. This will provide a clear geographical context and enhance the regional relevance of the findings. Please expand the "Sampling and Identification of Isolates" section to include specifics about the regions in Chile where the isolates were collected. Additionally, provide more detail about the areas within the plant sampled, such as specific equipment, surfaces, or products. Please emphasize the broader implications of studying L. monocytogenes in Chile in Introduction.
The authors only need to provide full names when they’re mentioned for the first time, such as for WGS (Line 24: the efficacy of WGS, Line 127: 2.3. WGS, Line 128: For WGS, and Line 177: for WGS data).
Line 32-33: ISL and cn should not be in italic.
Line 33-35: WGS has already been known for years for its usefulness in such studies. Therefore, this should not be the significance of your study. You should focus on your major findings instead. Conclusions should also be rewritten.
Line 139: NextSeq 2000
Line 144: Settings for SPAdes should be provided.
Line 442: Reference are normally not cited in Conclusions.
Author Response
Comments 1: The manuscript should highlight in the title, abstract, and relevant sections that the isolates were obtained from Chile. This will provide a clear geographical context and enhance the regional relevance of the findings.
Response 1: We agree with this comment. Therefore, we have added that the isolates were obtained in Chile in the title and abstract, along with the details of the source of the isolates.
Page 1; L3 and L4
L26
Table 2: L409
Comments 2: Please expand the "Sampling and Identification of Isolates" section to include specifics about the regions in Chile where the isolates were collected. Additionally, provide more detail about the areas within the plant sampled, such as specific equipment, surfaces, or products.
Response 2: Thank you to the reviewer for the suggestion. Additionally, we have rewritten section 2.1. Sampling and Identification of Isolates.
Page 3; Line 139 to 145.
New text:
After detecting a strain of L. monocytogenes in a refrigerated ready-to-eat (RTE) food, a comprehensive study was conducted at the RTE production plant located in Santiago, Chile (33°26′16″S, 70°39′01″W), to identify specific sources of contamination. A two-class sampling plan was applied with the parameters n=5; c=0; m=0; and M=0. A total of 216 samples were analyzed from various areas of the plant, including surfaces (floors, grates, and drains), equipment (worktables, wheels, and preparation tanks), raw materials, and finished products.
Comments 3: Please emphasize the broader implications of studying L. monocytogenes in Chile in Introduction
Response 3: Thank you to the reviewer for the suggestion. The following paragraph is added:
In Chile, two significant listeriosis outbreaks were reported between 2008 and 2009, with 165 and 73 cases, respectively. These outbreaks were associated with the consumption of goat cheese, sausages, and other meat products. The most frequently identified serotypes were 4b (CC1) and 1/2a (CC9) [Paduro et al., 2020].
Page 2; Line 75 to 78.
Comments 4: The authors only need to provide full names when they’re mentioned for the first time, such as for WGS (Line 24: the efficacy of WGS, Line 127: 2.3. WGS, Line 128: For WGS, and Line 177: for WGS data).
Response 4: Thank you for pointing this out, we have corrected WGS as indicated by the reviewer.
Line 25; L112; L194; L244; L265
Comments 5: Line 32-33: ISL and cn should not be in italic.
Response 5: Thank you for the suggestion: We have removed the italics from ISL and cn.
Line 33
Comments 6: Line 33-35: WGS has already been known for years for its usefulness in such studies. Therefore, this should not be the significance of your study. You should focus on your major findings instead. Conclusions should also be rewritten.
Response 6: Thank you to the reviewer for the suggestion. We have rewritten the conclusion of the abstract and the overall conclusion.
Abstract:
The presence of various antibiotic resistance genes, as well as resistance to thermal shocks and disinfectants, could confer the L. monocytogenes ST5 strains with acquired environmental resistance to the hygiene treatments used in the studied food production plant.
Line 34 to 36.
Conclusion:
The genomic analysis of the L. monocytogenes strains studied revealed the presence of various antibiotic resistance genes, as well as resistance to thermal shocks and disinfectants. These traits may confer acquired environmental resistance to the hygiene treatments used in the food production plant under study, thereby promoting the persistence of these L. monocytogenes ST5 strains. Consequently, the use of whole genome sequencing (WGS) provides valuable information that allows for more precise targeting of preventive sanitization programs, thus ensuring food safety and contributing to a healthier and safer diet.
Line 660 to 666
Comments 7: Line 139: NextSeq 2000
Response 7: Thank you to the reviewer for the suggestion. We have corrected "NextSeq 2000."
Line 186
Comments 8: Line 144: Settings for SPAdes should be provided.
Response 8: Thank you to the reviewer for the suggestion. We have corrected the phrase, as the adjustments to SPAdes were in the following sentence.
Subsequently, the reads were assembled using SPAdes v3.11.1 [24], where the contigs were filtered based on a minimum coverage of 5x and a minimum length of 200 bp using SeqSphere+ software v10.0.
Line 231 to 234
Comments 9: Line 442: Reference are normally not cited in Conclusions
Response 9: Thank you for pointing this out. We have removed the reference from the conclusions.
Line 666
Reviewer 2 Report
Comments and Suggestions for Authors
Listeria monocytogenes is a dangerous pathogen, the source of which for humans can be food, especially the RTE type. These rods adapt well to variable and unfavorable environmental conditions, including the ability to grow at low temperatures. Understanding the ecology of L. monocytogenes is essential to maintaining food safety.
Below are comments to the Authors:
- Title - L. monocytogenes please put in italics and please add isolated from Refrigerated Foods
Abstract:
- line 24 -add the number of strains that were sequenced
- line 28 - please correct to include serotype in one sentence and ST in the next sentence
Introduction:
- add brief information about listeriosis outbreaks linked to RTEs
- lines 86 i 87 - please use abbreviations
Material and methods:
- line 109 - italics
- line 128 - please use abbreviations
Results:
- line 209 - please correct editorially
- line 211 - imp gene ?
Discussion:
- line 303 - L. monocytogenes
- e.g. lines 320, 433, 439 - please use abbreviation
Author Response
Comments 1: - Title - L. monocytogenes please put in italics and please add isolated from Refrigerated Foods
Response 1: We appreciate this comment from the reviewer and have included it in the title: isolated from Refrigerated Foods
Line 3 to 4
Comments 2: - line 24 -add the number of strains that were sequenced
Response 2: We appreciate this comment from the reviewer and have added the 16 sequenced strains.
Line 25
Comments 3: - line 28 - please correct to include serotype in one sentence and ST in the next sentence
Response 3: We appreciate this comment from the reviewer, and the serotype and ST have been added in the following sentence.
Line 28 to 29
Comments 4: - add brief information about listeriosis outbreaks linked to RTEs
Response 4: We appreciate this comment from the reviewer, and the following paragraph has been added:
Between 2005 and 2020, a total of 3,628 cases of listeriosis were reported across 127 outbreaks associated with ready-to-eat (RTE) foods. Of these outbreaks, 54% occurred in the European region and 38% in the Americas. Additionally, 31% of the outbreaks were linked to RTE meat products, 28% to RTE dairy products, 13% to fresh or minimally processed RTE fruit and vegetable products, 12% to fish and seafood products, and the remainder to foods containing multiple ingredients [WHO, 2022].
Line 69 to 75
Comments 5: - lines 86 i 87 - please use abbreviations
Response 5: We thank the reviewer for this comment, and abbreviations have been used.
Line103 and Line 104
Comments 6: - line 109 – italics
Response 6: We thank the reviewer for this comment, and L. monocytogenes has been written in italics.
L177
Comments 7: - line 128 - please use abbreviations
Response 7: We appreciate this comment from the reviewer, and abbreviations have been used.
Line 194
Comments 8: - line 209 - please correct editorially
Response 8: We appreciate this comment from the reviewer, and the wording has been corrected.
Line 389-390
Comments 9: - line 211 - imp gene ?
Response 9: We appreciate this comment from the reviewer. Due to an unintentional error, the name of the gene was written incorrectly. It has been corrected to "iap gene."
Line 390
Comments 10: - line 303 - L. monocytogenes
Response 10: We appreciate this comment from the reviewer, and italics have been added.
Line 500
Comments 11: - e.g. lines 320, 433, 439 - please use abbreviation
Response 11: We appreciate this comment from the reviewer, and abbreviations have been used.
Line 510; Line 517; Line 676
Reviewer 3 Report
Comments and Suggestions for Authors
The aim of study was to evaluate the efficacy of WGS in the typing of Listeria monocytogenes isolated from refrigerated foods, compared to traditional typing.
The manuscript is well written, but in my opinion 16 strains isolated from the same plant are too few to reach significant conclusions.
The same study conducted on isolates from different plants would be very interesting, as the strains have been molecularly characterized in very great detail.
In my opinion, the work should be completely revised and restructured, focusing on the investigation carried out on the plant due to the positivity for L. monocytogenes found, describing in detail the sampling plan and then reporting the molecular characterization of the 16 strains isolated from the positive samples.
Author Response
Comments 1: The manuscript is well written, but in my opinion 16 strains isolated from the same plant are too few to reach significant conclusions.
Response 1: We appreciate the reviewer’s observation that the manuscript is well-written. In this context, and regarding the reviewer’s comments, we respectfully believe that the sampling conducted, in which 16 different strains of L. monocytogenes were isolated and initially typed by PFGE as highly similar, allows us to conclude the effectiveness and advantages of using WGS for the typing of this pathogen. The in-depth characterization of L. monocytogenes and the information generated provide an opportunity for the implementation of WGS in food production facilities, which could facilitate adjustments to sanitization plans, enhance the efficiency of L. monocytogenes control, and consequently contribute to safer food production.
Comments 2: The same study conducted on isolates from different plants would be very interesting, as the strains have been molecularly characterized in very great detail.
Response 2: Thank you for pointing this out, and we appreciate the reviewer for this comment, which we agree with; however, it is not the objective of our study.
Comments 3: In my opinion, the work should be completely revised and restructured, focusing on the investigation carried out on the plant due to the positivity for L. monocytogenes found, describing in detail the sampling plan and then reporting the molecular characterization of the 16 strains isolated from the positive samples.
Response 3: We appreciate the reviewer’s comment. In response, we have revised the entire manuscript, adding new paragraphs and conclusions that address, in part, the points raised. Additionally, this manuscript was developed in the context of a comparative analysis between PFGE and WGS in a food production plant, highlighting how the use of WGS can significantly contribute to the improvement of sanitation plans.
In this context, this work was prepared as a contribution to the Special Issue for which we were invited: "New Advances in Management and Characterization of Zoonotic Pathogens in Foodstuffs and Food Processing Facilities", considering that the scope of the Special Issue states:
"The topic of this Special Issue is to collect relevant papers that are able to shed light on the different aspects of the behavior of zoonotic pathogens in food processing environments and in foodstuffs, including the new advances in strain characterization, management of food processing environment contaminations, shelf-life studies, growth, and inactivation dynamics. Recently, omic technologies have been used to guarantee food authenticity and transparency, traceability, food safety investigation, and other fields."
Therefore, our manuscript contributes in this regard by highlighting the advantages of WGS, a technology that, despite its potential, is still not widely adopted due to its cost.
Line 672-678
Moreover, we have added details about the sampling plan, including a 2-class plan: n=5; c=0; m=0; M=0.
Line 139 to 145
Round 2
Reviewer 1 Report
Comments and Suggestions for Authors
Again, the authors only need to provide full names when they’re mentioned for the first time, such as for Line 107: in a refrigerated RTE. Please make sure you correct these problems throughout the manuscript.
Line 110: The authors should clarify the definitions of n, c, m, and M.
Line 172: 30-fold
Please correct all grammatical errors and typos in the manuscript.
Author Response
Comments 1: Again, the authors only need to provide full names when they’re mentioned for the first time, such as for Line 107: in a refrigerated RTE. Please make sure you correct these problems throughout the manuscript.
Response 1: Thank you for pointing this out. We have corrected it in the text.
L82
L119
L157
L174
L264
L270
L281
L288
L552
L568
L575
L581
L610
L621
L740
L791
Comments 2: Line 110: The authors should clarify the definitions of n, c, m, and M.
Response 2: Thank you for pointing this out. The concepts n, c, m, and M are internationally recognized. We have added a description of them in the text, which states: Where n is the number of sample units drawn; c: maximum allowable number of positive results; m: unacceptable tested samples; M: indicates the boundary between marginally acceptable counts and unacceptable counts.
L160 to L163
Comments 3: Line 172: 30-fold
Response 3: Thank you for pointing this out. It is corrected from 30 -fold to 30-fold.
L261
Comments 4: Please correct all grammatical errors and typos in the manuscript.
Response 4: Thank you for pointing this out. The manuscript has been reviewed again
L98
L264
L621
L740
Reviewer 3 Report
Comments and Suggestions for Authors
Suitable changes have been made, I have just an observation to make as regards the Conclusion section.
In my opinion, a short sentence on the advantages of WGS compared to traditional typing should be added, to better comply with the purpose of the special issue, as also highlighted by the authors.
Author Response
Comments 1: In my opinion, a short sentence on the advantages of WGS compared to traditional typing should be added, to better comply with the purpose of the special issue, as also highlighted by the authors.
Response 1: Thank you for pointing this out. We have added the following paragraph
In this context, WGS provides greater resolution and accuracy in strain identification compared to traditional typing, in addition to a genomic characterization which enhances the understanding of evolutionary relationships and responses in epidemiolog-ical surveillance. Accurate strain identification is key in the management of pathogens such as L. monocytogenes in food and processing facilities, as it is essential for ensuring food safety and protecting public health.
L864 to 870.